# Fighting COVID-19 Contagion among University Students of Healthcare Professions: An Italian Cross-Sectional Study

**DOI:** 10.3390/ijerph182312676

**Published:** 2021-12-01

**Authors:** Marco Tofani, Anna Berardi, Maurizio Marceca, Donatella Valente, Alfonso Mazzaccara, Antonella Polimeni, Giovanni Galeoto

**Affiliations:** 1Department of Public Health and Infectious Diseases, Sapienza University of Rome, 00185 Rome, Italy; maurizio.marceca@uniroma1.it; 2Department of Intensive Neurorehabilitation and Robotics, Bambino Gesù Children’s Hospital, 00165 Rome, Italy; 3Department of Human Neurosciences, Sapienza University of Rome, 00185 Rome, Italy; anna.berardi@uniroma1.it (A.B.); donatella.valente@uniroma1.it (D.V.); 4Istituto Neurologico Mediterraneo Neuromed Neuromed IRCCS, 86077 Pozzilli, Italy; 5Istituto Superiore di Sanità, 00185 Rome, Italy; alfonso.mazzaccara@iss.it; 6Department of Odontostomatological and Maxillo Facial Sciences, Sapienza University, 00185 Rome, Italy; antonella.polimeni@uniroma1.it

**Keywords:** COVID-19 prevention, distance learning, health professions, students, university

## Abstract

During the pandemic, most governments around the world temporarily closed educational institutions to contain the spread of the Coronavirus Disease 2019 (COVID-19). The objective of the present study is to evaluate the efficacy of an e-learning course on COVID-19 transmission for healthcare university students, in order to advance the preparedness of healthcare university students against contracting COVID-19 within the general university population. The e-learning course was run using a free web service for education. Access to the course was limited to participants enrolled in degree courses related to healthcare professions within the Italian university system. A specific and validated questionnaire was administered at two different times (pre-test and post-test). A paired sample t-test was then used to evaluate their knowledge on COVID-19. Furthermore, a questionnaire measuring their satisfaction was distributed. Data were analyzed from a qualitative point of view. The course was made available from March to July 2020. Over 25,000 students from different Italian universities and various backgrounds participated in the course. The analysis of final test scores revealed that approximately 97% of participants acquired new knowledge and skills on COVID-19, with a statistically significant improvement (*p* < 0.05). Therefore, it is possible to state that most students enrolled in degrees relating to healthcare at Italian universities are adequately trained with respect to COVID-19 knowledge. Furthermore, students declared a high satisfaction rate both with the course content, and with the management of the telematic platform used.

## 1. Introduction

Since the emergence of the novel coronavirus (SARS-CoV-2) infection in Wuhan, China, in December 2019, the epidemic has spread rapidly across countries worldwide [1]. On 30 January 2020, the World Health Organization (WHO) declared COVID-19 as a public health emergency of international concern and, finally, a pandemic in early March 2020 [2]. 

Individuals with COVID-19 can present with an influenza-like illness and respiratory tract infection demonstrating fever, cough, fatigue and/or shortness of breath. The spectrum of disease severity ranges from asymptomatic infection or mild upper respiratory tract illness to severe viral pneumonia with respiratory failure and/or death [3]. Furthermore, a majority of published studies to date have found that 50–70% of hospitalized patients exhibit several post-COVID symptoms up to 3 months after hospital discharge, such as brain fog, loss of attention, chest pain, general fatigue, dyspnea, cough, throat pain, myalgias, anxiety, depression and insomnia [4].

Worldwide, extensive measures to reduce person-to-person transmission of COVID-19 are required to control the current outbreak. Efforts for protecting and reducing contagion among susceptible populations, such as people with disabilities, the elderly and healthcare professionals were applied. Different countries are continuing to implement national and subnational measures for responding to the COVID-19 public health crisis. Many governments, at all levels, have reacted quickly by proposing national or local lockdowns, including educational institutions. Nevertheless, in Italy, many healthcare students attended university hospitals during the pre-pandemic situation. They continued to support healthcare professionals during the emergency both in university hospitals, and COVID-19 referred hospitals.

The Decree of the President of the Council of Ministers, 8 March 2020, containing further provisions implementing the decree-law of 23 February 2020, in No. 6 on urgent containment and management measures of the epidemiological emergency from COVID-19, suspended educational activities in all school levels, including universities. However, courses for resident medical doctors and specific training courses in general medicine, and the activities of trainees in the health professions should have continued [5].

On 11 March 2020, the WHO officially declared a pandemic situation for COVID-19 [6]. On 28 February 2020, within the Italian regulations on continuous medical education, the Italian National Institute of Health (ISS) created a training course for healthcare professionals on the emergency due to SARS-CoV-2. The course was based on available scientific evidence and official sources of information and updates [7]. Experts with different backgrounds developed the course: a group of ISS experts created the scientific content while experts on e-learning methods created the course format. Considering that Italian university students of healthcare continued to perform internships during the pandemic, it became crucial to offer comprehensive knowledge about controlling COVID-19 contagion. Therefore, this study aims to improve the knowledge and skills acquired by students in healthcare at Italian universities through an e-learning course on COVID-19. The secondary objective is to evaluate the quality of the course as perceived by the students through a satisfaction questionnaire.

## 2. Methods

On 26 March 2020, the research team, which already had experience managing online courses [8,9,10,11], was commissioned by the President of ISS to deliver the course. Researchers and healthcare professionals of the Sapienza University of Rome, together with the Rehabilitation and Outcome Measures Assessment (ROMA) Association, a non-profit organization, performed the management of the web platform.

### 2.1. The Training Course

The course, “Health emergency from new coronavirus SARS-CoV-2: preparation and contrast,” was developed by the ISS experts. The general objective was to allow health professionals to appropriately deal with the health emergency due to the new coronavirus SARS-CoV-2 by using available scientific evidence and official sources of information and updates. The course ran for a duration of 16 h and was to conclude within four weeks of enrollment. Other than frontal lectures, the teaching method used was active, asynchronous, with low interaction and inspired by the principles of problem-based learning (PBL), in which individual participants are activated by defining and achieving their objectives. In doing so, participants acquire new elements of knowledge and new skills for solving the problem itself [12]. The course was structured into three learning units:Characteristics of the international and national health emergency situation due to SARS-CoV-2;Surveillance, detection, and management of suspected cases;Information for health care providers for prevention, identification, and control in clinical settings.

### 2.2. Course Management

The e-learning course on COVID-19 transmission course was made available using a free web service for education. The research group decided to use the “Google Classroom” platform as it met the criteria of affordability, ease of use and dissemination capacity. The e-learning training course created by ISS was then uploaded to the platform.

As a first step, a specific email address (corsonuovocoronavirus@gmail.com) was created to administer the courses and provide adequate support to students. Because the platform allows the creation of a class hosting a maximum of 250 students, 120 twins’ classes were generated. For each of the 120 classes, a specific code was automatically generated from Google Classroom to access the class. These codes were then published online and constantly updated using a specific webpage of the ROMA Association, partner of the project. In addition, the ROMA Association offered a help desk service for students who experienced difficulties accessing the course and other technical queries. Furthermore, at the end of the course, once participants answered both the post-test and satisfactory survey, each student obtained a certificate of completion. ROMA association, together with partners, sent certificates to the students.

### 2.3. Participants

Students were recruited in all Italian universities in collaboration with the Conferenza Permanente delle Classi di Laurea delle Professioni Sanitarie (National Committee of Health Professions). The research group sent the e-course information and Participants’ Guide to the committee, where the committee’s board then delivered the guideline to the Presidents and Didactic Directors of the Degree Courses of the Italian Universities, who then dispersed it to their students. To access the course, participants had to be enrolled in the Degree Courses of the Health Professions of Italian Universities. Participation in the course was voluntary and free of charge for the university, the course of study and the students. At the time of enrollment, participants were informed of the methods and objectives of the project. An initial test and a final test were digitally administered to all students. The test results were communicated individually to the participants, while the aggregate data were transmitted to the coordinators of the courses of study for each university.

### 2.4. Assessment Tools and Analysis

For assessing skills and knowledge, a questionnaire was administered at two different times (pre-and post-test). The questionnaire was developed and validated by the research group, during a previous validation study in a sub-population of healthcare students in Rome, Italy; psychometric properties were investigated in terms of internal consistency, responsiveness, sensitivity and specificity [13]. The test consisted of 31 multiple choice questions with 4 answer options where only 1 was correct. The questionnaire had to be completed at the commencement and the end of the course. To obtain a certificate of completion, the minimum score was set equal to or greater than 23; cut-off was determined with a previous validation study, using the ROC curve [13]. For the evaluation of the satisfaction on the course, a 5-point Likert questionnaire was used. The survey consisted of 18 questions, and the answers were rated from 1 (completely disagree) to 5 (completely agree).

To analyze data, a descriptive analysis was performed for sampling, and percentages of students who passed the course, and the response rates for satisfaction were assessed. In addition, a Student’s t-test for paired samples was used to assess knowledge improvement through pre-and post-test scores: statistical significance was set for *p* < 0.05. All statistical analysis was performed using IBM SPSS version 23.00 (IBM Corp., Armonk, NY, USA).

## 3. Results

From March to July 2020, 25,479 students from the healthcare professions courses of Italian Universities participated in the course. Of these students, 1172 (4.6%) continued their internship activities during the training course period, and of these 28% had contact with a COVID-19 positive patient. 74% of people who attended the course were female, 55.8% were students in Nursing and Midwifery health professions and more than half of the population were enrolled in universities of Italy’s central regions. Sample characteristics are presented in Table 1.

From the evaluation through Student’s t-test for paired samples, a statistically significant improvement in student competencies was obtained in all degree courses (*p* < 0.05). Results are summarized in Table 2.

From the evaluation of the sample, it is possible to observe that only 3.4% did not pass the course, obtaining a score below 23 on the final test. Results are summarized in Table 3.

The evaluation of the responses to the satisfaction questionnaire shows that evaluation/judgement of all questions received a score on average of greater than 4 (response range from 1 to 5). Table 4 shows results more in-depth.

## 4. Discussion

The emergency caused by the propagation of COVID-19 has led the National Government to adopt measures to control contagion, where restrictions also involved higher education and professional training [12]. While most university students used distance learning approaches, many healthcare students continued to attend hospitals and healthcare centers [14]. Other than Italy, in other European countries, medical and nursing students remained willing to care for patients with COVID-19; however, lack of knowledge about basic measures to prevent the transmission of this virus at both community and hospital levels, and the low percentage of students who reported having received specific training, are striking [15]. Adequate awareness of COVID-19 in the healthcare setting is required, and different studies highlight the importance of education and training for those populations [16,17,18]. Different studies highlighted the need of COVID-19 education for medical doctors and healthcare professions students. It has been observed that the majority of students acquired information on COVID-19 from social media which is an inauthentic source of obtaining evidence about diseases [19]. In addition, official sites such as the CDC website, and medical search engines such as PubMed, which should reflect reliable sources of information, were less commonly used than social media and news channels to obtain information [20]. Health literacy and knowledge were found to protect medical students from fear [21], and to produce positive attitudes and perceptions about COVID-19 [22].

To the best of our knowledge, the present research report results from one of the first mass online training courses on COVID-19 for students of different healthcare professions, including rehabilitation and technical professions.

The total number of students who attended the training course was 25,749, amounting to approximately 30% of the Italian students of healthcare professions [23]. More than 90% of the students are enrolled within the first three years of their degree, because healthcare professions in Italy are organized in three years of academic degree coursework. The majority of the students (51.3%) came from central Italy. This is likely due to two main factors: firstly, in central Italy—particularly in the city of Rome—are situated the largest universities in Italy in terms of student population and course availability; secondly, since the research group is also based in central Italy, the probability and opportunity to reach out to students for the study were increased. Residents in medicine who participated in the study made up only a fraction (0.7%) of total participants. This finding was expected as more medical residents, as health professionals, were involved in an official course provided by the National Institute of Health (NIH). Nevertheless, some medical residents discovered information about attending the course on the internet and decided to participate. This result confirms the high attraction of the course for Italian university students.

The analysis of the final test scores revealed that approximately 97% of the participants acquired new knowledge and skills in relation to COVID-19. This result is significant for several reasons: firstly, 4.6% of students declared they attended hospitals even during the emergency phase; secondly, as healthcare professional students were more susceptible to bear close contact with COVID-19 affected patients, this likely instilled students with greater confidence concerning COVID-19 prevention. All healthcare professional students saw a homogeneous improvement, and no significant differences between classes were found. The fact that there are no differences between the samples emphasizes that no specific knowledge was needed in advance to pass the test. The student satisfaction survey additionally confirmed this. Excellent results emerged from completed student satisfaction surveys; all questions received an average score evaluation/judgement greater than 4. Considering that the maximum score was 5, the results highlight a high level of student satisfaction.

Despite these encouraging results, the research group desires to highlight some limits and challenges in delivering the course. First, the entire project was organized without ad hoc funding. Consequently, it is possible to set-up a large online course within a limited resource setting. Certainly, the absence of ad hoc funding influenced the choice of the e-learning platform, and the possibility to increase efficiency. Using Google Classroom obligated students to either have or create a Google account. Although this may be not entirely ethically appropriate, the working group opted for this strategy given the service was free of charge, and the students were not required to contribute to the costs in any way. Furthermore, the choice to use a free e-learning platform obligated the research group to create 120 twin classes and reduced transfer of certificates and communications to universities. However, the partnership with a non-profit organization proved immensely helpful for managing access codes, aiding communication to the students and producing web certificates.

Furthermore, considering that students voluntarily decided to participate in the course, a possible selection bias can be observed. Indeed, trained students might be more responsive to this topic, and the greater responsiveness might be a predictive indicator of learning aptitude and greater knowledge.

Ultimately, it is important to stress that vaccination campaigns are the first method to counteract the COVID-19 pandemic; however, sufficient vaccination coverage is conditioned by the public’s knowledge and acceptance of these vaccines. The study was conducted during the first wave of the pandemic, when vaccines were not yet available, and the epidemiological situation was different. Although high levels of COVID-19 vaccination acceptance and knowledge in Italian undergraduates have been reported, it has also been demonstrated that information and education strategies require on-going and/or constant monitoring [24]. Therefore, supplementary e-learning courses on this issue must be proposed and their efficacy evaluated.

## 5. Conclusions

It is possible to state that a significant part of the health profession students at Italian universities were successfully trained. The students declared themselves to be highly satisfied with both the course contents and platform management. It is hoped that this work will prove helpful to other universities, including those in low- and middle-income countries that require training in fighting COVID-19 contagion.

## Figures and Tables

**Table 1 ijerph-18-12676-t001:** Sample characteristics (total 25,479).

Sample Characteristics (Total 25,479)	Absolute Values	Percentage Values
**Age** (mean ± SD)	22.78 ± 4.7	
**Sex (female)**	18,790	73.7
**Courses typology**		
Nursing and Midwifery Health Professions (BSc)	14,227	55.8
Rehabilitation Health Professions	5446	21.4
Technical and Diagnostic Health Professions	3913	15.4
Prevention Health Professions	1289	5.1
MSc in Rehabilitation Sciences	432	1.7
Medical Degree Course	170	0.7
**University Location**		
North Regions	8121	31.9
Central Regions	13,073	51.3
Southern Regions	4285	16.8
**Year of Course**		
I	8162	32.0
II	8109	31.8
III	7942	31.2
IV	77	0.3
V	28	0.1
VI	36	0.1
Outsite prescribed time	1125	4.4
**Students who attended clinical practice**		
Yes	1172	4.6
No	24,307	95.4
*** Students who had a contact with COVID-19 patient**		
Yes	331	28.2
No	841	71.8

* Total number of students (1172).

**Table 2 ijerph-18-12676-t002:** Pre-post evaluation (Student’s t-test for paired samples).

Degree Courses	Pre-Test Mean ± SD	Post-Test Mean ± SD	t	*p*
Nursing and Midwifery Health Professions (BSc)	24.53 ± 4.069	28.44 ± 2.888	−90.507	0.001 *
Rehabilitation Health Professions	24.52 ± 4.033	28.98 ± 2.471	−66.725	0.001 *
Technical and Diagnostic Health Professions	24.48 ± 3.918	28.63 ± 2.654	−54.192	0.001 *
Prevention Health Professions	24.39 ± 4.013	28.68 ± 2.758	−29.978	0.001 *
MSc for Health in Rehabilitation Professions	24.85 ± 4.216	28.45 ± 2.612	−14.587	0.001 *
Medical Degree Course	25.15 ± 4.597	28.92 ± 2.983	−79.38	0.001 *

* *p* < 0.05.

**Table 3 ijerph-18-12676-t003:** Percentage of students who passed the test.

Degree Courses	Test Passed *n*(%)	Test Failed *n*(%)
Nursing and Midwifery Health Professions	13,646 (95.9)	581 (4.1)
Rehabilitation Health Professions	5339 (98.0)	107 (2.0)
Technical and Diagnostic Health Professions	3870 (98.9)	49 (1.1)
Prevention Health Professions	1182 (91.7)	107 (8.3)
MSc in Rehab Sciences	419 (97.0)	13 (3.0)
Medical Degree Course	166 (97.6)	4 (2.4)
**Total**	24,618 (96.6)	861 (3.4)

**Table 4 ijerph-18-12676-t004:** Results of satisfaction questionnaire.

Questions	1	2	3	4	5	Mean ± SD
1.1 The course objectives were clear (%)	0.1	0.6	7.7	41.7	49.9	4.41 ± 0.67
1.2 Contents were consistent with the course objectives (%)	0.1	0.6	6.1	37.5	55.7	4.48 ± 0.65
1.3 The teaching methodology was effective (%)	0.5	2.6	15.0	43.1	38.8	4.17 ± 0.81
1.4 The level of education was adequate to my knowledge (%)	0.2	1.3	13.3	46.0	39.1	4.23 ± 0.74
1.5 I learned new concepts (%)	0.3	1.1	7.6	34.6	56.4	4.46 ± 0.71
1.6 I have acquired new skills (%)	0.8	2.9	16.9	41.0	38.4	4.13 ± 0.85
1.7 I can transfer what I have learned to my internship (%)	0.7	2.0	13.1	38.0	46.2	4.27 ± 0.81
2.1 The available study materials were adequate to acquire necessary information (%)	0.3	1.0	9.5	41.9	47.3	4.35 ± 0.72
2.2 The quality of available study materials was appropriate (%)	0.3	1.2	9.9	42.6	46.0	4.33 ± 0.73
2.3 The available study materials were up-to-date with respect to the most recent literature (%)	0.4	1.4	10.8	41.3	46.2	4.31 ± 0.75
2.4 The overall organization (modules, timing, pre and final evaluations) was satisfactory (%)	0.4	1.7	12.1	43.6	42.2	4.25 ± 0.76
2.5 Consultation of the Participant’s Guide was helpful in guiding me through the course. (%)	0.5	2.3	18.8	43.5	34.9	4.10 ± 0.81
2.6 Test questions were sufficiently clear (%)	0.3	1.7	11.1	42.0	44.9	4.29 ± 0.76
2.7 The time available to perform the tests was adequate (%)	0.1	0.5	5.4	31.0	63.0	4.56 ± 0.63
3.1 The quality of technical support for the course was satisfactory (%)	0.3	1.3	14.2	43.7	40.5	4.23 ± 0.76
3.2 The quality of mentoring for this course was satisfactory (%)	0.4	1.7	17.7	43.9	36.4	4.14 ± 0.79
4.1 The operation of web platform for education was adequate (%)	0.7	2.6	11.9	39.3	45.6	4.27 ± 0.82
4.2 Ways to access the platform was simple and immediate (%)	1.1	3.5	12.8	34.3	48.3	4.25 ± 0.89

## Data Availability

Data are available from the corresponding author upon reasonable request.

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
