# Peer review of "Fighting COVID-19 Contagion among University Students of Healthcare Professions: An Italian Cross-Sectional Study"

_ijerph, 2021, doi:10.3390/ijerph182312676_

Round 1
Reviewer 1 Report
The article presents the results of an online course set up for health students during the pandemic period, to give them detailed information on COVID-19.The course was broadcast to all Italian universities, enrolling 25,479 students, which represents a major study.
The objective of the study is to improve the content of the course, and to have feedback on student satisfaction. The process is well explained, the elements of analysis well detailed. However, some points are not fully explained: - why do the majority of enrolled students come from universities in central Italy? - the results of the “prevention health professions” test are less good than for the other categories. Explanation ? - has the information given in the course changed over time? If so, what is the impact on the results? Table 1: typology, not tipology The article is well written, easy to read.
Author Response
Point 1: why do the majority of enrolled students come from universities in central Italy?
Response 1: Thank you for your observation. The majority of the students come from universities in central Italy for two main reasons: first, in the central Italy especially in Rome there are the biggest universities in Italy (as example Sapienza University of Rome has the number of students of healthcare professions of four or five University in the North). The second reason is probably due to the fact that the research group is located in central Italy, so we had a greater capacity for recruitment in our area. We specified it in Discussion section, please see lines 195 - 199
Point 2: the results of the “prevention health professions” test are less good than for the other categories. Explanation ?
Response 2: We think is a good question and as research group we have discussed this issue, unfortunately without finding any valid hypothesis. We apologize for this answer.
Point 3: has the information given in the course changed over time? If so, what is the impact on the results?
Response 3: thank you for your question. The recruitment period ranging from March to July. Although knowledge about COVID-19 was increasing, we did not have enough evidences to change some classes in the course or for new recommendations. Of course, these emerged in the next months, and the Italian National Institute of Health modified the course according to the available evidences.
Point 4: Table 1: typology, not tipology Response 4: Thank you
The article is well written, easy to read.

Reviewer 2 Report
I had the pleasure of reviewing the article entitled: “Fighting covid-19 contagion among university students of healthcare professions: an Italian cross-sectional study”.
The aim of this study is to evaluate the efficacy of an e-learning course on COVID-19 transmission for healthcare university students in order to advance the preparedness of healthcare university students against contrasting COVID-19 within the general university population.
The article is very interesting, well written and easy to read. The introduction is clearly written, the whole is supported by properly selected literature. It is exhaustive in each of its sections and deals with a very interesting topic. I especially emphasize the importance and contribution of the previously validated questionnaire that was used during the examination of the acquired knowledge of students and after the e-learning course (pre-test and post-test).In the current crisis of the Covid-19 pandemic, this work will be of great help to other universities to provide appropriate much-needed training to students in the fight against Covid-19 infection, especially given the appropriate teaching and valorisation methodology described inthis paper.
I suggest that you study the following references (their topic is related to the research topic you conducted) in order to complete the chapter of discussions in your paper:
Haque A, Mumtaz S, Khattak O, Mumtaz R, Ahmed A. Comparing the preventive behavior of medical students and physicians in the era of COVID-19: Novel medical problems demand novel curricular interventions. Biochem Mol Biol Educ. 2020;10.
Khasawneh AI, Humeidan AA, Alsulaiman JW, et al. Medical Students and COVID-19: Knowledge, Attitudes, and Precautionary Measures. A Descriptive Study From Jordan. Front Public Health. 2020;8:253.
Ikhlaq A, Bint-E-Riaz H, Bashir I, Ijaz F. Awareness and Attitude of Undergraduate Medical Students towards 2019-novel Corona virus. Pak J Med Sci. 2020;36(COVID19-S4):S32-S36.
Gohel KH, Patel PB, Shah PM, Patel JR, Pandit N, Raut A. Knowledge and perceptions about COVID-19 among the medical and allied health science students in India: An online cross-sectional surve. Clin Epidemiol Glob Health. 2020;10.
Long N, Wolpaw DR, Boothe D, et al. Contributions of Health Professions Students to Health System Needs During the COVID-19 Pandemic: Potential Strategies and Process for U.S. Medical Schools. Acad Med. 2020;10.
Hagana A, Cecula P. Medical Students in the Time of COVID-19: Opportunities and Challenges. AEM Educ Train. 2020;4(3):291.
Çalışkan F, Mıdık Ö, Baykan Z, et al. The knowledge level and perceptions toward COVID-19 among Turkish final year medical students. Postgrad Med. 2020;1-9.
Gao Z, Ying S, Liu J, Zhang H, Li J, Ma C. A cross-sectional study: Comparing the attitude and knowledge of medical and non-medical students toward 2019 novel coronavirus. J Infect Public Health. 2020.
Rzymski P, Nowicki M. COVID-19-related prejudice toward Asian medical students: A consequence of SARS-CoV-2 fears in Poland. J Infect Public Health. 2020;13(6):873-876.
Nguyen HT, Do BN, Pham KM, et al. Fear of COVID-19 Scale-Associations of Its Scores with Health Literacy and Health-Related Behaviors among Medical Students. Int J Environ Res Public Health. 2020;17(11):4164.
Hjiej G, Fourtassi M. Medical students' dilemma during the Covid-19 pandemic; between the will to help and the fear of contamination. Med Educ Online. 2020;25(1):1784374.
Khamees D, Brown CA, Arribas M, Murphey AC, Haas MRC, House JB. In Crisis: Medical Students in the COVID-19 Pandemic. AEM Educ Train. 2020;4(3):284-290.
Before accepting a paper for publication, it is necessary to make corrections in terms of editing the text in accordance with the rules of the journal. It is also necessary to edit in detail the cited references throughout the text of the paper and in reference chapter in accordance with the rules of the journal.
Author Response
Point 1: I suggest that you study the following references (their topic is related to the research topic you conducted) in order to complete the chapter of discussions in your paper
Response 1: Thank you vary much for the references. We integrate in the discussion of the paper. Please see discussion section
Point 2: Before accepting a paper for publication, it is necessary to make corrections in terms of editing the text in accordance with the rules of the journal. It is also necessary to edit in detail the cited references throughout the text of the paper and in reference chapter in accordance with the rules of the journal.
Response 2: Thank you. We checked all the manuscript according rules of the journal. We hope it is now good.

Reviewer 3 Report
Good morning authors,
I think that your paper is so interesting. I congrulate you for this. However, I write you some recommendations:
- In table 4, there are another font type. You must improved it.
- I think that the bibliography can be improved and more extensive. I recommend you that write about de professors training (and the importance of this) in the introduction and discussion.
- Now, there are a lot of papers about teaching in times of crisis. For this reason, you can talk about it in the introduction and discussion. You can use:
del Arco, I.; Silva, P.; Flores, O. (2021). University Teaching in Time of Confinement: The light and shadows of compulsory online learning. Sustainability, 13(1), 375. https://doi.org/10.3390/su13010375
- In the introduction, you talk about the inconveniences in health of COVID-19. You can expand this section.
Author Response
Point 1: In table 4, there are another font type. You must improved it. Response 1: Thank you, we modified
Point 2: I think that the bibliography can be improved and more extensive. I recommend you that write about de professors training (and the importance of this) in the introduction and discussion. Now, there are a lot of papers about teaching in times of crisis. For this reason, you can talk about it in the introduction and discussion. You can use: del Arco, I.; Silva, P.; Flores, O. (2021). University Teaching in Time of Confinement: The light and shadows of compulsory online learning. Sustainability, 13(1), 375. https://doi.org/10.3390/su13010375
Response 2: We agreed with you. Thank you for your suggestion. We added different research papers in the references and better discussed the Discussion Section. Please see page 6 lines 178 - 186
Point 3: In the introduction, you talk about the inconveniences in health of COVID-19. You can expand this section.
Response 3: Thank you for your suggestion. As recommended we expanded introduction section talking about inconveniences in health of COVID. Please see Page 1 lines 37-44
